# Diversity and Evolution of Viral Pathogen Community in Cave Nectar Bats (*Eonycteris spelaea*)

**DOI:** 10.3390/v11030250

**Published:** 2019-03-12

**Authors:** Ian H Mendenhall, Dolyce Low Hong Wen, Jayanthi Jayakumar, Vithiagaran Gunalan, Linfa Wang, Sebastian Mauer-Stroh, Yvonne C.F. Su, Gavin J.D. Smith

**Affiliations:** 1Programme in Emerging Infectious Diseases, Duke-NUS Medical School, Singapore 169857, Singapore; dolyce.low@u.nus.edu (D.L.H.W.); jayanthi.jayakumar@duke-nus.edu.sg (J.J.); linfa.wang@duke-nus.edu.sg (L.W.); yvonne.su@duke-nus.edu.sg (Y.C.F.S.) gavin.smith@duke-nus.edu.sg (G.J.D.S.); 2NUS Graduate School for Integrative Sciences and Engineering, National University of Singapore, Singapore 119077, Singapore; 3Bioinformatics Institute, Agency for Science, Technology and Research, Singapore 138671, Singapore; vithiagarang@bii.a-star.edu.sg (V.G.); sebastianms@bii.a-star.edu.sg (S.M.-S.); 4Department of Biological Sciences, National University of Singapore, Singapore 117558, Singapore; 5SingHealth Duke-NUS Global Health Institute, SingHealth Duke-NUS Academic Medical Centre, Singapore 168753, Singapore; 6Duke Global Health Institute, Duke University, Durham, NC 27710, USA

**Keywords:** Metaviromics, Southeast Asia, adenovirus, bunyavirus, flavivirus, herpesvirus, papillomavirus, paramyxovirus, parvovirus, picornavirus, polyomavirus, poxvirus, reovirus, rotavirus

## Abstract

Bats are unique mammals, exhibit distinctive life history traits and have unique immunological approaches to suppression of viral diseases upon infection. High-throughput next-generation sequencing has been used in characterizing the virome of different bat species. The cave nectar bat, *Eonycteris spelaea*, has a broad geographical range across Southeast Asia, India and southern China, however, little is known about their involvement in virus transmission. Here we investigate the diversity and abundance of viral communities from a colony of *Eonycteris spelaea* residing in Singapore. Our results detected 47 and 22 different virus families from bat fecal and urine samples, respectively. Among these, we identify a large number of virus families including *Adenoviridae, Flaviviridae, Reoviridae, Papillomaviridae, Paramyxoviridae, Parvoviridae, Picornaviridae,* and *Polyomaviridae.* In most cases, viral sequences from *Eonycteris spelaea* are genetically related to a group of bat viruses from other bat genera (e.g., *Eidolon*, *Miniopterus*, *Rhinolophus* and *Rousettus*). The results of this study improve our knowledge of the host range, spread and evolution of several important viral pathogens. More significantly, our findings provide a baseline to study the temporal patterns of virus shedding and how they correlate with bat phenological trends.

## 1. Introduction

The advent of next generation sequencing (NGS) technologies has drastically increased the discovery of novel viruses and estimates of virus diversity [1,2]. Though family-level specific primers are often used to screen diagnostic samples, they are designed based on available reference sequences and typically target the most conserved genetic region such as the polymerase genes [3,4,5]. These assays often lack sensitivity at the expense of detecting an entire family of viruses. Next generation sequencing can detect viruses at low concentrations and often provides sequence reads from across the entire genome, providing sites for primer walking and gap closing [6,7,8]. This approach can also detect divergent lineages that may not be amplified using traditional polymerase chain reaction (PCR) approaches [9]. However, as host and bacterial components can dominate sequencing reads, virus reads of interest tend to be in low abundance in these data sets [10].

NGS has been employed to detect zoonotic pathogens and in numerous cases of virus discovery and metavirome descriptions, including bats in China, Myanmar, New Zealand and North America [11,12,13,14,15]. This technique has also detected novel poxviruses and adenoviruses, divergent papillomaviruses, and unique paramyxoviruses [16,17,18]. More recently, genetic regions from a novel filovirus were identified from *Rousettus* bat in China in 2015, indicating the utility of this approach to ascertain the presence of potentially pathogenic viruses in reservoir hosts [19].

Increasingly, these studies have focused on bats because this group of animals are unique virus reservoirs. Bats are distinctive mammals, having ecological, immunological and behavioral attributes that set them apart from other orders. Bats are exceptionally speciose, comprising 20% of all mammalian species and are the only mammals that are capable of true flight [20]. Many species are gregarious and roost in large colonies, which can number over one million individuals [21]. They are relatively long-lived for their body size and temperate species often undergo torpor or hibernation [22]. There are several theories regarding why bats are exceptional viral reservoirs and rarely experience pathogenesis with infection. One is that they have to deal with the physiological stress of flight, with increased metabolic rates and a subsequent increase in reactive oxygen species [23,24]. Recent research demonstrated that the unique innate immune system of bats may allow them to co-exist with viruses, maintaining very low levels of viremia or keeping viruses in a quiescent state [25,26,27].

*Eonycteris spelaea* is a nectivorous bat with a distribution ranging from the Malay–Indonesian archipelago to southern China, extending west into the Indian subcontinent [28]. These cave-roosting bats are especially important pollinators of durian [29]. There are two known colonies of *E. spelaea* in Singapore and both populations are restricted to roosting under bridges. This species forage at distances greater than 30 km from their roosting site, but are threatened across their range by habitat loss and hunting for human consumption [30]. Here we perform next generation sequencing on pooled feces and pooled urine samples collected from one colony of *E. spelaea* to identify the viral diversity and to examine the difference in virus communities between fecal and urine samples. Furthermore, we conducted phylogenetic analysis to understand the evolutionary relationships of different families of viruses detected from this study.

## 2. Material and Methods

### 2.1. Sample Collection for NGS Library Preparation

Urine and fecal samples were collected from 3 time points from a colony of the cave nectar bats (*Eonycteris spelaea*) in Singapore for NGS. Feces were collected on 14 March, 28 March and 11 April 2013 while urine was collected on 24 April, 8 May and 20 May 2014. Disposable plastic drop cloths were placed under the colony and approximately 25 g of feces was collected and placed in a 50-mL tube. Urine samples from approximately 100 bats was placed into viral transport media (penicillin, streptomycin, polymyxin B, gentamicin, nystatin, olfoxacin, sulfamethoxazole). Fecal material was prepared for library preparation as previously described [31] while urine was centrifuged at 10,000 × g for 3 min to pellet debris. RNA-zol was added to urine viral supernatant and RNA extracted using Direct-zol™ RNA MiniPrep (Zymo Research Corporation, Irvine, CA, USA), then subjected to in-column DNase I digestion (New England BioLabs Inc., Ipswich, MA, USA). Extracted and DNase I digested RNA was treated with Ribo-Zero™ Gold rRNA Removal Kit (Epidemiology) (Epicentre, Madison, WI, USA) and 500 bp cDNA libraries were constructed using NEBNext^®^ Ultra™ Directional RNA Library Prep Kit for Illumina^®^ (New England BioLabs Inc., Ipswich, MA, USA) and visualized on 1.5% agarose gel, before being excised and purified using Zymoclean Gel DNA Recovery Kit (Zymo Research Corporation, Irvine, CA, USA). All sequencing reactions were run at the Duke-NUS Genome Biology Facility. Libraries from urine were run on an Illumina MiSeq machine with paired ends and a read length of 2 × 250 bp, while fecal sample libraries were processed as described previously and run on an Illumina HiSeq 2000 with paired ends and a read length of 2 × 76bp. 

### 2.2. NGS Data Analysis

All FASTQ files were assessed using FastQC to assess overall quality [32]. Trimming was executed using Trimmomatic-0.3.2 to remove adapters, low quality bases (Q = 20 with a sliding window 4) and reads with fewer than 50 bp length [33]. Taxonomic read classification was performed with DIAMOND sequence similarity searches against a local National Center for Biotechnology Information (NCBI) nr (non-redundant) protein database [34]. DIAMOND outputs were imported into MEGAN6 for the taxonomic binning of reads and visualizing the distribution of virus family reads [35]. X174 phage reads were removed from the final data set as these are spiked-in as control for the next generation sequencing reaction. Virus family reads were exported for phylogenetic analysis. To confirm whether the reads sorted by MEGAN were true positives, these were verified as viral hits using the BLASTX tool from the National Center of Biotechnology Information.

### 2.3. Sample Collection and PCR Assays for Detection of Specific Viruses

Individual urine samples were collected from 59 time points and pooled by date (2014-03-27 to 2016-09-01). Pooled *Eonycteris* urine samples were centrifuged at 10,000 × *g* for 1 min. RNA was extracted from the supernatant using QIAamp^®^ Viral RNA Mini Kit (#52906; Qiagen Duesseldorf, Germany) and cDNA made with Random Hexamers and SuperScript^®^ II Reverse Transcriptase (#18064-014, Invitrogen, CA, USA). This was used to screen for orthoreovirus and paramyxovirus RNA-dependent RNA polymerase (RdRp) gene with family-specific primers [5,36]. PCR products were visualized on a 1.5% agarose gel, and 500 bp bands from the paramyxovirus assay and 240 bp bands from the orthoreovirus assay were excised, gel purified with Qiagen QIAquick Gel Purification kit and sent for Sanger sequencing with both forward and reverse reads sequenced (1^st^ Base DNA Sequencing Services, Axil Scientific Pte Ltd., Singapore).

### 2.4. Phylogenetic Analysis

Candidate reads from mammalian-specific viruses were *de novo* assembled in Geneious 7.1.6 (Biomatters Ltd., Auckland, New Zealand) and consensus sequences were used for subsequent phylogenetic analysis [37]. Representative nucleotide sequences specific to the gene of interest were downloaded from the NCBI GenBank for the following virus families (Table 1): *Adenoviridae, Flaviviridae, Reoviridae, Papillomaviridae, Paramyxoviridae, Parvoviridae, Picornaviridae,* and *Polyomaviridae*. For each viral family, individual sequence data sets were aligned using Transalign [38] and MAFFT [39] followed by manual curation of alignments. Gene phylogenies were initially reconstructed using FastTree [40], and the final data sets were further down sampled to reduce redundant and similar sequences. Altogether, 15 individual data sets were analyzed based on the following viral genes: adenovirus polymerase, flavivirus envelope, flavivirus NS5, paramyxovirus nucleoprotein, paramyxovirus polymerase, parvovirus VP1, parvovirus VP2, picornavirus 3D, picornavirus polyprotein, orthoreovirus M2, orthoreovirus L2, rotavirus VP1, rotavirus VP7, polyomavirus VP2, and 36 papillomavirus E1 (Table 1). Individual gene phylogenies were reconstructed using RAxML [41] with a general time reversible model and robustness of the nodes was assessed using 1000 bootstrap replicates. 

## 3. Results

### 3.1. Next Generation Sequencing Analysis

Four NGS data sets generated from the pooled urine and fecal samples: two libraries (Urine-25 and Urine-27) were constructed from pooled urine that were run on Illumina MiSeq, while two other libraries (Feces-MiSeq and Feces-HiSeq) from pooled feces were sequenced using Illumina MiSeq and HiSeq, respectively. For urine samples, a total of 5,126,632 reads from Urine-25 and 13,421,263 reads from Urine-27 were generated, and approximately 29.8% and 30.8% of respective reads could be assigned to known sequences in the GenBank nr database by DIAMOND (Table 2). For the fecal data set, a total of 68,584,413 reads from the Feces-HiSeq and 4,952,973 reads from the Feces-MiSeq were generated, but the percentage of assigned reads from the data sets differed dramatically (5.8% from Feces-HiSeq and 75.7% from Feces-MiSeq). The majority of assigned reads in the fecal and urine samples were from Eukaryota. Reads assigned to bacteria amounted to between 11.2–14.6% of the four NGS data sets. Fungal sequences were rare in the urine data sets, but common in the fecal data sets, comprising 16.9% of assigned reads in Feces-HiSeq and 35.4% of assigned reads in Feces-MiSeq. Viral reads were a relatively small component, totaling approximately 1% of all assigned reads from the four data sets. All NGS data sets are available at the National Center for Biotechnology Information Sequence Read Archive under bioProject PRJNA524946.

Our results detected 22 and 47 different virus families from urine and fecal samples of *Eonycteris spelaea*, respectively (Table 3). We observed differences in viral presence and absence between urine and fecal samples. There were 25 virus families were unique to the fecal data sets and seven of these are known to infect vertebrates: *Bunyaviridae*, *Caliciviridae*, *Papillomaviridae*, *Parvoviridae*, *Picobirniviridae*, *Picornaviridae*, and *Rhabdoviridae*. Moreover, the most common virus families found in the urine are retroviruses (25.38% of total reads), dicistroviruses (20.59%), and coronaviruses (15.22%) (Table 3). The fecal samples virus reads were primarily dicistroviruses (58.1%), which are arthropod-specific viruses with phenotypes ranging from asymptomatic infection to high mortality in insects [42]. Viruses in the families *Siphoviridae* (14.7%) and *Podoviridae* (15.4%), both bacteriophages, were commonly identified. Other viruses detected in the bat fecal samples included parvoviruses (0.54%), retroviruses (0.99%), reoviruses (0.15%), picornaviruses (0.14%), and polyomaviruses (0.14%) (Table 3). For the pooled urine screening from 2014-03-27 to 2016-09-01, a total of 15 samples (25.4%) were positive for paramyxoviruses, while 1 sample (1.7%) was positive for orthoreoviruses. 

### 3.2. Adenoviridae

Of the 360 total adenovirus reads from the fecal and urine data sets, there were 39 contigs assembled with 17 reads unassembled. The longest contig was 539 bp, while the shortest was 115 bp. A 504 bp contig was similar to the mamadenovirus polymerase genes. The polymerase phylogeny (Appendix A) of adenovirus clearly indicates *Eonycteris* adenovirus is well nested within a monophyletic clade (72% bootstrap support, BS) of bat adenovirus from *Rousettus* and *Minopterus* species. The new adenovirus sequence from Singapore (MK603133) is most closely related to two *Rousettus leschenaultii* adenovirus sequences (KX961095 and KX961096) from China in 2013. They share a high level of nucleotide (nt: 86.6%) and amino acid (aa: 99.4%) similarities.

### 3.3. Flaviviridae

Two contigs were assembled from 62 flavivirus reads from the fecal and urine data sets. These were mapped to an envelope (E) protein and a NS5 protein, which encodes methyltransferase and RNA-dependent RNA polymerase, respectively. The E phylogeny (Appendix A) of Flavivirus indicates that the *Eonycteris* sequence (MK603134) closely resembles Phnom Penh virus from a *Cynopterus* bat (NC_034007), with 88.4% nucleotide (nt) similarity and 92.7% amino acid (aa) similarity. They formed a strongly supported monophyletic clade (100% BS) with Batu Cave virus from Malaysia (KJ469370). Similar observations are found in the NS5 phylogeny (Appendix A), although the *Eonycteris* sequence (MK603135) appears to be more closely related to Batu Cave virus (58% BS; nt: 92.5%, aa: 100% similarity). 

### 3.4. Reoviridae

The 172 reovirus reads from the fecal data set were assembled into 41 contigs with 23 unassembled reads. The longest contig was 487 bp, while the shortest was 76 bp. There were members of the genera *Orthoreovirus* and *Rotavirus* in the data set. Four contigs were selected for phylogenetic reconstruction. Two contigs (206 bp and 199bp) were similar to the orthoreovirus M2 gene, which encodes the viral outer capsid proteins (ς1 and μ1c) involved in receptor binding and host cell membrane penetration. The M2 (624bp) phylogeny (Figure 1) of orthoreovirus revealed the *Eonycteris* sequence (MK603137) is most closely related to Melaka virus from Malaysia (JF342664) with 98.5% amino acid (aa) similarity and 97.2% nucleotide (nt) similarity. We also detected a 318-bp contig of the L1 gene of reovirus sequences. This gene encodes the RNA dependent RNA polymerase. The *Eonycteris* L1 sequence (MK603136) (Figure 1) is most closely related to Cangyuan orthoreovirus (KM382260) and Melaka orthoreovirus (JF342661), both with 100% aa similarity and 97.5% nt similarity. The *Eoncyteris* L1 segment also formed a strongly supported monophyletic clade (100% BS) with other bat ortheroviruses, including; Melaka virus, Cangyuan virus (KM382260), Kampar virus (JF342655), Pulau virus (JF342667), Nelson Bay reovirus (JF342673) and Pteropine reovirus (KM279381). 

The other *Orthoreovirus* contigs belong to the genus Rotavirus, with a 321 bp contig similar to the VP1 gene or RNA-dependent RNA polymerase (RdRp) and the second contig (291 bp) mapping to the outer capsid protein sequence (VP7). The VP1 gene (Figure 2) of *Eoncyteris* rotavirus was most similar to rotavirus detected in a *Rousettus leschenaultii* from China in 2005 (KX814935), with an 76.6% aa similarity and 73.5% nt similarity. The *Eonycteris* and *Rousettus* sequences (MK603148) formed a well-supported monophyletic clade (99% BS) (Figure 2). In contrast, the Rotavirus VP7 gene sequence from *Eonycteris* (MK603149) forms a polytomy with other rotavirus sequences from a broad range of hosts including humans, pigs, bovines and equines (Figure 2).

### 3.5. Papillomaviridae

There were 12 papillomavirus reads from the fecal data set and 9 were assembled to produce 3 contigs (L1 protein and two E1 protein contigs), with a maximum length of 635 bp and a minimum length of 77 bp. One contig (439 bp) was similar to the V3 gene which encodes the minor capsid protein. The papillomavirus E1 phylogeny (Appendix A) indicates the *Eonycteris* sequence (MK603138) is most similar to an *Eidolon helvum* papillomavirus from Cameroon (KX276956). These two sequences have 68.4% aa similarity and 69% nt similarity. These two sequences were in a monophyletic group with another *Eidolon helvum* papillomavirus from Cameroon (KX276957) (100% BS).

### 3.6. Paramyxoviridae

There were 57 reads from the fecal and urine data sets were classified as from *Paramyxoviridae* of which 56 were assembled to produce 6 contigs. The longest contig was 338 bp and the shortest was 127 bp and there were sequences from the polymerase gene (L) and nucleocapsid (NP). From the family-specific PCR, there were two unique polymerase sequences generated (471 bp) with 61.8% aa similarity and 64.3% nt similarity. The two L-gene contigs of *Eonycteris* (MK603139, MK603140) (Figure 3) were closely related to those from *Eidolon helvum* paramyxoviruses found in Ghana and Republic of the Congo. In comparison, two NP sequences (MK603141 and MK603142) (Figure 3) from *Eonycteris* (98.4% aa similarity) are in a sister group to human henipaviruses and other bat borne paramyxoviruses (74% BS). 

### 3.7. Parvoviridae

There were 565 *Parvovirinae* reads from the fecal data sets. These produced 43 contigs with a maximum length of 891 bp and a minimum length of 120 bp. Two contigs matched with the capsid gene (VP1 and VP2) of parvovirus. The VP1 contig (321 bp) from *Eonycteris* (MK603143) is related to a *Rousettus leschenaultii* parvovirus from in China (100% BS) (MF682925), and they shared a 73.5% nt similarity and 76.6% aa similarity. These two sequences also formed a strongly supported monophyletic clade with porcine bocaviruses (97% BS) (Figure 4). Similar to the VP1 gene, the VP2 sequence (MK603144) (319 bp) from *Eonycteris* is sister to several porcine bocaviruses, forming a well-supported clade (87% BS) (Figure 4).

### 3.8. Picornaviridae

There were 159 reads from the fecal data sets classified in *Picornaviridae*. These created 33 contigs, the longest at 724 bp and the shortest 76 bp. One contig was from the polyprotein region and picornavirus phylogeny (Appendix A) indicates the *Eonycteris* sequence (MK603145) was sister to a monophyletic clade (100% BS) comprising three bat picornaviruses (KJ641693, HQ595345, NC_015934) from *Rhinolophus hipposideros*, *R. sinicus* and *Hipposideros armiger* from China and Hong Kong. These shared a 68.4% nt similarity and a 51.7% aa similarity. Another read matched the 3D gene that encodes for RNA polymerase. The 3D gene from *Eonycteris* (MK603146) (Appendix A) was related to bat picornavirus from *Vespertilio superans* and *Myotis altarium* from China (72.8% nt and 79.2% aa similarity), although this node was not statistically supported.

### 3.9. Polyomaviridae

A total of 164 reads from the fecal and urine data sets were assigned to the virus family *Polyomaviridae* and 33 contigs were assembled. The shortest contig was 89 bp and the longest was 525 bp. One read of 420 bp was from the VP2 major capsid gene. The VP2 of polyomavirus phylogeny (Appendix A) indicates the *Eonycteris* sequence (MK603147) forms a strongly supported monophyletic clade (100% BS) with a bat polyomavirus from *Hipposideros pomona* from China (nt: 91.7% similarity and aa: 97.1% similarity). In addition, it is apparent that polyomavirus is capable of infecting diverse species of different bat families, including Miniopteridae (e.g., *Miniopterus* and *Myotis*), Molossidae (e.g., *Otomops*), Pteropodidae (e.g., *Acerodon*, *Dohsonia*, *Eidolon*, *Rousettus*), Phyllostomidae (e.g., *Artibeus*), and Rhinolophidae (e.g., *Rhinolopus*).

### 3.10. Additional Viruses Detected

In addition to the above-mentioned viruses identified from fecal and urine samples of *Eonycteris* bats in Singapore, there were other virus families with a low number of reads (<15). For instance, one read (114 bp) was identified as a bunyavirus sequence that is most similar to the L protein of phlebovirus found in blacklegged ticks. Three reads corresponded to Taterapox virus (Poxviridae), originally isolated from an African gerbil and 15 reads for herpesviruses. Phylogenetic relationships of these viral pathogens were not reconstructed due to a lack of reference sequences in GenBank.

Notably, insect viruses and plant/fungal viruses were frequently identified in our bat samples from Singapore. The most common invertebrate viruses were dicistroviruses (65,353 reads), but there were several other virus families present (*Alphatetraviridae*, *Balculoviridae*, *Carmotetraviridae*, *Caulimoviridae*, *Nodaviridae*, *Nudiviridae*, *Permutotetraviridae* and *Polydnaviridae*). Moreover, detectable plant/fungal viruses include *Alphaflexiviridae*, *Betaflexiviridae*, *Edornaviridae*, *Luteoviridae*, *Narnaviridae*, *Partitiviridae*, *Phycodnaviridae*, *Potyviridae*, *Secoviridae*, *Tymoviridae* and *Virgaviridae*. Taken together, our study identifies a broad diversity of viruses present in *Eonycteris* bats in Singapore.

## 4. Discussion

This comparative metagenomic study reports the fecal and urine virome of the cave nectar bat, *Eonycteris spelaea*, in Singapore. Our NGS findings detected a broad diversity of viral pathogens present in *Eonycteris* bat species, with 43 and 22 different virus families from the fecal and urine samples, respectively. Among these, we identified a large number of virus families that commonly infect vertebrates, including *Adenoviridae, Flaviviridae, Reoviridae, Papillomaviridae, Paramyxoviridae, Parvoviridae, Picornaviridae* and *Polyomaviridae.* Previous virus surveillance on *Eonycteris spelaea* detected genomic evidence of astroviruses, coronaviruses and filoviruses with serologic evidence of filoviruses from Singapore and China [31,43,44]. In other Southeast Asian sampling sites, there is genomic evidence of flaviviruses (Phnom Penh bat virus) and bunyaviruses (Issyk-kul-Keterah virus), while there is serologic evidence of Nipah virus antibodies in *E. spelaea* from Malaysia [45,46,47]. Previous metagenomic studies on bats have revealed several novel viruses from a diverse group of viral families [13,16,18,48,49]. The virus families discovered in this study have been detected in several species of bats by conventional PCR and next generation sequencing, including adenoviruses [16,50,51], bunyaviruses [52,53,54], flaviviruses [55,56,57], herpesviruses [16,49,58,59], paramyxoviruses [60,61,62], papillomaviruses [14,16,63,64], parvoviruses [12,59,65,66], picornaviruses [15,59,67], polyomaviruses [14,68,69,70], poxviruses [16,71,72] and reoviruses [73,74,75,76]. 

We reconstructed individual gene phylogenies for each of the above-mentioned virus families. In most phylogenies, our novel viral sequences of *Eonycteris spelaea* are clustered within a group of bat viruses from other genera, often from the same family (*Pteropodidae*), indicating circulation of these viruses among different bat species. Detection of novel viruses may not provide direct information on zoonotic capacity, but the generated sequence data can allow us to better understand the evolutionary history of these virus families and infer potential cross-species transmission [77]. For instance, our VP1 phylogeny of parvovirus indicates that bat parvoviruses are sister to murine and porcine parvoviruses, whereas the VP2 phylogeny shows bat parvoviruses are closely related to canine, feline and porcine parvoviruses. It is apparent that parvovirus is capable of infecting a broad host species, although further research is needed to understand how the virus jumps to different hosts. 

Viruses exhibit specific tissue tropisms based on available cellular receptors and compatibility of the intracellular environment. Detection of these viruses depends on what tissue or sample type is being screened. Previous studies have indicated which viruses are more likely to be shed in feces (adenoviruses, astroviruses, parvoviruses, picornaviruses), urine (paramyxovirues), or tissues [60,78,79,80,81]. Receptors can be widely available, such as in paramyxovirus (CD46 in measles and sialic acid in Sendai virus) and picornavirus infections (ICAM-1 in Coxsackie virus), or narrowly restricted in adenovirus infections (integrin on monocytes) [82]. Due to limited commercial reagents and the difficulty of maintaining experimental colonies, little work has been done to characterize receptors and viral tissue preference in bats, though progress is being made in determining coronavirus tropism [83,84,85].

Consensus family level primers are used globally for virus biosurveillance and amplify the most conserved genomic region, usually the polymerase, but these primers may be of limited utility in detecting divergent strains [86]. Unbiased NGS reads will capture reads scattered across the virus genome, depending on the starting quantity of the sample and virus [10]. As sequencing reactions generate reads from all nucleic acids, reads can end up binned as unknown because there are no similar reads available in reference data sets (GenBank nucleotide and non-redundant RefSeq proteins). To minimize having unassigned reads, our approach used the program DIAMOND and the nr data set to only assign sequences that have homology to coding regions, eliminating false assignments of ribosomal RNA [34]. Our NGS sequencing of bat fecal and urine samples resulted in a low proportion of viral reads. This is common in metagenomic data sets where the majority of assigned reads were from the host, bacteria and viruses that infect plants, bacteria and insects [14]. These large data sets often provide low coverage across the genomes of viruses due to the host and bacterial background [87]. Interestingly, in our study, more viral reads were generated from the MiSeq Illumina than the HiSeq Illumina. Interestingly, retrovirus reads were much more common in the urine data set compared to the fecal data set. This may be caused by the comparatively low background in the urine. 

In this study, we characterized the fecal and urine virome of *Eonycteris spelaea*, an ecologically important species [30]. The NGS data sets provided sequencing reads for 10 families of viruses that are known to infect mammals, providing segments to develop strain-specific assays to detect and quantify these viruses for future longitudinal studies. Recent research has demonstrated that co-roosting and colony size may be important in the generation of novel variants and in viral maintenance [88,89]. As this species roosts in large numbers, co-roosts with several other species of bats, and is widely distributed, it may be a candidate species to understand if virus diversity is based on phylogenetic relatedness, co-roosting partners or geographic separation. Additionally, with the sequencing of the full genome and the presence of a captive colony of bats at Duke-NUS Medical School, this will expand our capacity in understanding the genomics, infection and immunology of host-virus interactions using *E. spelaea* as the studied species [90].

## Figures and Tables

**Figure 1 viruses-11-00250-f001:**
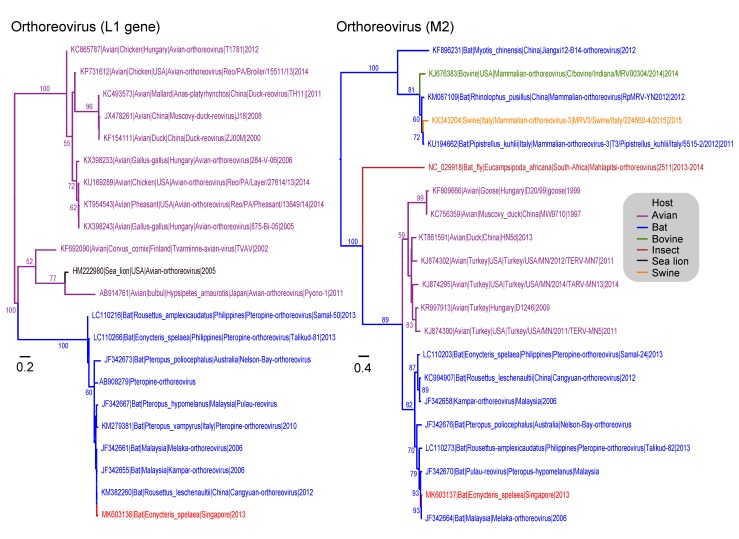
Phylogenetic relationships of the L1 and M2 gene sequences of orthoreovirus, inferred by using the maximum-likelihood method with the generalised time-reversible (GTR) + GAMMA distribution model in RAxML. Colored branches represent viruses isolated from different hosts. Red branches denote new sequences collected from *Eonycteris spelaea* bats in Singapore. Bootstrap support values greater than 50% are displayed at major nodes. The scale bar indicates the number of nucleotide substitutions per site.

**Figure 2 viruses-11-00250-f002:**
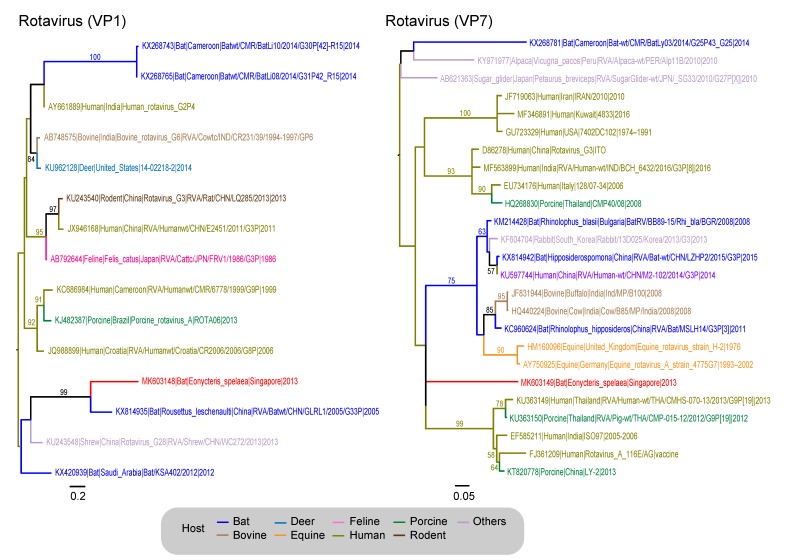
Phylogenetic relationships of the VP1 and VP7 gene sequences of rotavirus, inferred by using the maximum-likelihood method with the GTR + GAMMA model in RAxML. Colored branches represent viruses isolated from different hosts. Red branches denote new sequences collected from *Eonycteris spelaea* bats in Singapore. Bootstrap support values greater than 50% are displayed at major nodes. The scale bar indicates the number of nucleotide substitutions per site.

**Figure 3 viruses-11-00250-f003:**
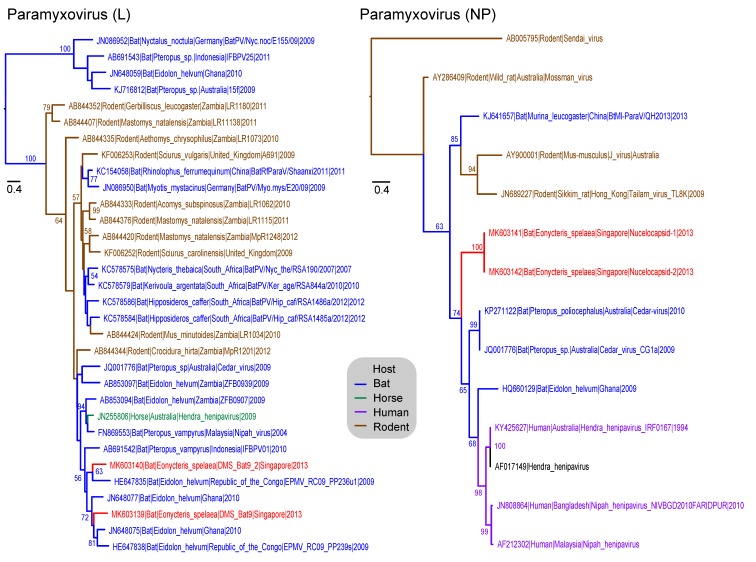
Phylogenetic relationships of the L and NP gene sequences of paramyxovirus, inferred by using the maximum-likelihood method with the GTR + GAMMA model in RAxML. Colored branches represent viruses isolated from different hosts. Red branches denote new sequences collected from *Eonycteris spelaea* bats in Singapore. Bootstrap support values greater than 50% are displayed at major nodes. The scale bar indicates the number of nucleotide substitutions per site.

**Figure 4 viruses-11-00250-f004:**
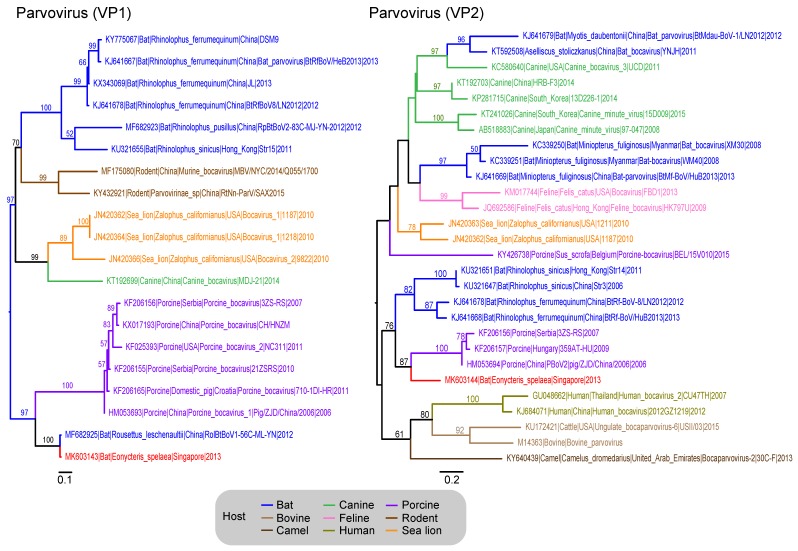
Phylogenetic relationships of the VP1 and VP2 gene sequences of parvovirus, inferred by using the maximum-likelihood method with the GTR + GAMMA model in RAxML. Colored branches represent viruses isolated from different hosts. Red branches denote new sequences collected from *Eonycteris spelaea* bats in Singapore. Bootstrap support values greater than 50% are displayed at major nodes. The scale bar indicates the number of nucleotide substitutions per site.

**Table 1 viruses-11-00250-t001:** Virus family alignments and sequence lengths.

Virus Family (Genus)	Gene	Initial Sequence Alignment	Down Sampled Sequence Alignment	Down Sampled Alignment Length (bp)
*Adenoviridae*	DNA Polymerase	227	37	885
*Flaviviridae*	E–Envelope	398	45	333
	NS5–Non-structural protein 5	600	47	231
*Papillomaviridae*	V3–minor capsid protein	368	36	969
*Paramyxoviridae*	N–Nucleoprotein	15	14	720
	L–Polymerase	743	32	471
*Parvoviridae*	VP1–Capsid	27	20	528
	VP2–Capsid	501	28	323
*Picornaviridae*	3D–RNA polymerase	1753	20	246
	Polyprotein	40	23	813
*Polyomaviridae*	E1–major capsid protein	289	26	441
*Reoviridae* (*Orthoreovirus*)	M2–viral outer capsid proteins (ς1 and μ1c)	149	21	624
	L2–core spike protein λ2	30	22	318
*Reoviridae* (*Rotavirus*)	VP1–RNA-dependent RNA polymerase	501	15	231
	VP7–outer capsid protein	536	25	291

**Table 2 viruses-11-00250-t002:** Next generation sequencing reads by data set and selected taxonomic ranks.

Data set-Name	Total Reads in Data set	Total Reads Assigned by MEGAN	Eukaryotes	Mammalia	Arthropoda	Bacteria	Archaea	Fungi	Virus
Urine-MiSeq-25	5,126,632	1,527,375	1,154,068	1,010,668	583	222,953	204	8217	1606
Urine-MiSeq-27	13,421,263	4,137,771	3,230,684	2,994,683	1201	548,911	497	20,496	3691
Feces HiSeq	68,584,413	3,993,465	2,178,408	69,136	53,546	546,757	68	675,967	21,856
Feces MiSeq	4,952,973	3,750,870	2,836,374	54,457	60,533	420,446	97	1,326,425	106,823

**Table 3 viruses-11-00250-t003:** Virus reads from next generation sequencing data set from *Eonycteris spelaea* urine and feces.

Virus Family	Urine-MiSeq-25 Reads	Urine-MiSeq-27 Reads	Urine reads by Family (% of Total)	Feces-Hiseq Reads	Feces-MiSeq Reads	Fecal Reads by Family (% of Total)	Total Reads	Consensus Reads/Unassembled Reads
*Adenoviridae* ^V^	171	115	286 (7.14%)	31	43	74 (0.07%)	360 (0.31%)	28/17
*Alphaflexiviridae* ^P/F^	-	-	-	-	143	143 (0.13%)	143 (0.12%)	28 / 13
*Alphatetraviridae* ^A^	-	-	-	-	1	1	1	-
*Astroviridae* ^V^	50	-	50 (1.25%)	6	8	14 (0.01%)	64 (0.05%)	6/3
*Baculoviridae* ^A^	-	-	-	1	2	3	3	0/3
*Betaflexiviridae* ^P/F^	4	-	4 (0.10%)	9	14	23 (0.02%)	27 (0.02%)	7/9
*Bunyaviridae* ^V^	-	-	-	-	1	1	1	-
*Caliciviridae* ^V^	-	-	-	-	9	9 (0.01%)	9 (0.01%)	4/1
*Carmotetraviridae* ^A^	-	-	-	-	34	34 (0.03%)	34 (0.03%)	5/0
*Caulimoviridae* ^A/P^	-	-	-	3	5	8 (0.01%)	8 (0.01%)	2/4
*Chrysoviridae* ^F^	15	-	15 (0.37%)	3	3	6 (0.01%)	21 (0.02%)	5/2
*Coronaviridae* ^V^	170	440	610 (15.22%)	46	87	133 (0.12%)	743 (0.64%)	52/21
*Dicistroviridae* ^A^	120	705	825 (20.59%)	292	65,061	65,353 (58.09%)	66,178 (56.80%)	2326/1546
*Endornaviridae* ^P/F^	-	-	-	-	4	4	4	2/0
*Flaviviridae* ^V^	10	52	62 (1.55%)	-	1	1	63 (0.05%)	2/1
*Herpesviridae* ^V^	25	-	25 (0.62%)	3	12	15 (0.01%)	40 (0.03%)	7/5
*Iflaviridae* ^A^	8	-	8 (0.20%)	46	578	624 (0.55%)	632 (0.54%)	79/21
*Inoviridae* ^B^	-	-	-	1	-	1	1	-
*Leviviridae* ^B^	-	-	-	-	4	4	4	2/0
*Luteoviridae* ^P^	-	-	-	-	1	1	1	-
*Microviridae* ^B^	17	117	134 (3.34%)	3016	438	3454 (3.07%)	3588 (3.08%)	39/8
*Mimiviridae* ^PR^	-	-	-	-	1	1	1	-
*Myoviridae* ^B^	1	-	1 (0.02%)	4029	1688	5717 (5.08%)	5718 (4.91%)	525/346
*Narnaviridae* ^F^	-	-	-	16	345	361 (0.32%)	361 (0.31%)	42/6
*Nodaviridae* ^A^	-	-	-	-	117	117 (0.10%)	117 (0.10%)	13/10
*Nudiviridae* ^A^	-	-	-	28	63	91 (0.08%)	91 (0.08%)	32/16
*Papillomaviridae* ^V^	-	-	-	3	9	12 (0.01%)	12 (0.01%)	3/3
*Paramyxoviridae* ^V^	14	34	48 (1.20%)	3	6	9 (0.01%)	57 (0.05%)	6/1
*Partitiviridae* ^P/F^	12	129	141 (3.52%)	12	22	34 (0.03%)	175 (0.15%)	12/11
*Parvoviridae* ^V^	-	-	-	203	421	624 (0.55%)	624 (0.54%)	55/21
Genus: *Parvovirinae*	-	-	-	187	370	557 (0.50%)	557 (0.48%)	-
Genus: *Denovirinae*	-	-	-	8	51	59 (0.05%)	59 (0.05%)	-
*Permutotetraviridae* ^A^	-	-	-	55	933	988 (0.88%)	988 (0.85%)	92/7
*Phycodnaviridae* ^P^	3	1	4 (0.10%)	-	2	2	6 (0.01%)	1/3
*Picobirnaviridae* ^V^	-	-	-	-	3	3	3	0/3
*Picornaviridae* ^V^	-	-	-	45	114	159 (0.14%)	159 (0.14%)	33/17
*Podoviridae* ^B^	26	-	26 (0.65%)	4002	13,349	17,351 (15.42%)	17,377 (14.91%)	787/225
*Polydnaviridae* ^A^	-	-	-	4	6	10 (0.01%)	10 (0.01%)	3/4
*Polyomaviridae* ^V^	8	-	8 (0.20%)	63	93	156 (0.14%)	164 (0.14%)	33/13
*Potyviridae* ^P^	-	-	-	44	15	59 (0.05%)	59 (0.05%)	13/12
*Poxviridae* ^V^	16	-	16 (0.40%)	2	1	3	19 (0.02%)	3/3
*Reoviridae* ^V^	1	-	1 (0.02%)	95	76	171 (0.15%)	172 (0.15%)	42/22
Genus: *Orthoreovirus*	1	-	1	53	25	78 (0.07%)	103 (0.09%)	-
Genus: *Rotavirus*	-	-	-	24	39	63 (0.06%)	102 (0.09%)	-
*Retroviridae* ^V^	552	465	1017 (25.38%)	95	39	134 (0.12%)	1151 (0.99%)	170/61
*Rhabdoviridae* ^A^	-	-	-	-	11	11 (0.01%)	11 (0.01%)	4/2
*Secoviridae* ^P^	-	87	87 (2.17%)	-	33	33 (0.03%)	120 (0.10%)	6/2
*Siphoviridae* ^B^	17	325	342 (8.54%)	8839	7662	16,501 (14.67%)	16,843 (14.46%)	518/144
*Totiviridae* ^V/A/P^	16	281	297 (7.41%)	-	47	47 (0.04%)	344 (0.30%)	23/12
*Tymoviridae* ^P^	-	-	-	1	8	9 (0.01%)	9 (0.01%)	3/1
*Virgaviridae* ^P^	-	-	-	-	2	2	2	1/0
Total Reads	1256	2751	4007		20,996		116,518	

V = Vertebrate; A = Arthropod; P = Plant; F = Fungi; B = Bacteria; PR = Protist.

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
