# Peer review of "Diversity and Evolution of Viral Pathogen Community in Cave Nectar Bats (Eonycteris spelaea)"

_viruses, 2019, doi:10.3390/v11030250_

Reviewer 1 Report

This paper describing the identification of viruses present in Cave Nectar bats (Eonycteris spelaea) through metagenomic sequencing, provides a generally well written, interesting preliminary study of the types of viruses that can be found in this species of bat. Specific comments are provided below. One general problem was the use of virus database sequences for phylogenetic comparison that were derived from different sets of virus isolates when reconstructing the phylogeny of two different genes from the same virus family. Ideally, all alignments and trees would be reconstructed from the same set of whole virus genome sequences for each gene in a family. In addition, the discussion should be carefully reviewed to improve grammar and clarity.

 ·     Proper italicization of virus taxon names needs to be used.

·     Lines 113-144 Methods: The reference to “RNA-dependent DNA polymerase (RdRp) gene…” should be “RNA-dependent RNA polymerase (RdRp) gene…”

·     Line 123 Methods: The indication that “consensus sequences were used” suggests that there was some variation among the assembled reads at individual genomic positions. Was this variation due to sequencing error or might it represent quasispecies virus populations or mixed populations of multiple distinct isolates?

·     Tables 3,4: It would be useful to know the number of distinct, nonoverlapping reads obtained for each family.

·     Results: The presentation of results for each detected virus family presumably covers reads from both the urine and fecal samples combined. This should be clearly stated. 

·     In Supplementary figure 2, the sequence datasets used for phylogenetic reconstruction of the flavivirus E and NS5 genes are different. In some cases, the gene sequences are extracted from whole genome sequences present in GenBank, and in others, they are derived from genome segments present in GenBank. But many of the genome segment sequences have whole genome equivalents present. For example, AF013394 (NS5, 1,011 nt) is 99% similar to NC_034007 (used for E), which is a whole genome sequence. The problem is that this makes it difficult to compare the two trees and detect incongruent relationships between each inferred phylogeny. This is true of all of the 2-tree figures as well. Perhaps the sequences derived from the same (or nearly the same) virus isolate could be labeled with a common identifier. In other cases, the gene segment corresponding to the same isolate was not used in constructing the tree. For Figure 2, rotavirus, the G33P rotavirus isolate sequence for VP1 (KX814935) was used in the alignment. The sequence for the same isolate representing the VP2 sequence was not included in the VP2 tree, even though it is available in GenBank (KX814936). The polytomy described for the VP2 tree (line 233) may very well have been resolved with the inclusion of KX814936. To the extent possible, sequences representing the same isolates should be used in reconstructing the phylogeny for two separate genes of the same virus species.

·      Lines 227-235. This paragraph references the rotavirus VP7 gene when it appears that the reference should be to the VP2 gene. It also references figure 4, when it should reference figure 2.

·     Figure 3, paramyxovirus trees: Why did the Eonycteris spelaea paramyxovirus sequences for NP contain “Nipah” in the label, and why were the L sequences differently labeled? Is there evidence that the sequence reads for these two paramyxovirus genes come from different species? 

·     Line 307: The phrase “but not limited to other virus families” is a bit awkward.

·     Discussion, line 346: The sentence beginning “Cross-species transmission is difficult…” is poorly worded and the mention of the human immune system seems out of place.

·     Line 360: The paragraph that begins here needs to be completely rethought and rewritten. It contains a mix of ideas and statements that may not necessarily provide the best explanations for the results obtained.

·     Line 381: The sentence beginning “As this species often exists in very large roosts…” is difficult to comprehend.

 Author Response

We would like to thank both reviewer for taking the time to review our manuscript and provide comments and suggestions to improve the quality. We have provided specific responses to each reviewer below. Reviewer one suggested that we provide the number of consensus sequences assembled from individual reads for each virus family. We revisited the NGS datasets and determined the number of consensus reads and unassembled reads for each virus. This lead us to combine tables 3 and 4 into one table that contained the virus families for feces and urine with the number of assembled reads as the last column. For trimming the data, we excluded any X174 phage reads as they likely originated from the library preparation kit. While reviewing the viruses by family in MEGAN from the DIAMOND file, it appeared that there were other bacteriophage sequences that had been removed during the analysis. We have included these reads in the table (families Microviridae, Myoviridae, and Podoviridae). Additionally, the reads from another bacteriophage family (Siphonviridae) were underestimated. These have been added to Table 3 (number of reads). While it does change the proportion of reads and reduces the percentage of reads from dicistroviruses (arthropod-specific viruses), the number of vertebrate-specific viruses remains unchanged. We apologize for this oversight and have remedied it in this revised version of the draft. We have uploaded our NGS fastq files to NCBI’s Sequence Read Archive so they will be publicly available. We have submitted our sequences used in the tree to Genbank and are awaiting accession numbers. Once these are accepted, we will add these to the manuscript and change the identifiers in the phylogenetic tree. Please note that our line references are based on the revised manuscript with changes being tracked.

 REVIEWER 1

 This paper describing the identification of viruses present in Cave Nectar bats (Eonycteris spelaea) through metagenomic sequencing, provides a generally well written, interesting preliminary study of the types of viruses that can be found in this species of bat. Specific comments are provided below. One general problem was the use of virus database sequences for phylogenetic comparison that were derived from different sets of virus isolates when reconstructing the phylogeny of two different genes from the same virus family. Ideally, all alignments and trees would be reconstructed from the same set of whole virus genome sequences for each gene in a family. In addition, the discussion should be carefully reviewed to improve grammar and clarity.

 We have addressed the specific comments below with our comments prefaced by ** and in blue

 **We thank the reviewer for their constructive comments. We agree that the optimal sequence alignments would use full genomes in both datasets where we have trees from multiple genes from the same virus family. With the majority of bat virus sequence discovery, there are several virus families where there are few full genomes available because the majority of reads generated are from family-specific or pan-viral primer sets. Full genomes are difficult to generate as they often don’t grow in commercial cell lines, laboratories may not have sufficient biosafety levels and primer walking can be difficult as the surface protein sequences are very divergent, making primer walking tough. Our objective for this was to determine the nearest neighbors/viruses to those detected in the Eonycteris feces and urine. We determined that the best resolution was from using sequences from several bat viruses, including those from fruit bats, to generate phylogenetic trees. We have reviewed the discussion and edited it for grammar and clarity.

 ·     Proper italicization of virus taxon names needs to be used.

**We have italicized the virus taxon names throughout the manuscript.

·     Lines 113-144 Methods: The reference to “RNA-dependent DNA polymerase (RdRp) gene…” should be “RNA-dependent RNA polymerase (RdRp) gene…”

**Line 188: We have changed the RNA-dependent DNA polymerase (RdRp) gene to RNA-dependent RNA polymerase (RdRp) gene

·     Line 123 Methods: The indication that “consensus sequences were used” suggests that there was some variation among the assembled reads at individual genomic positions. Was this variation due to sequencing error or might it represent quasispecies virus populations or mixed populations of multiple distinct isolates?

**The consensus sequences were derived from NGS reads (contigs) to generate overall longer reads for use in the phylogenetic reconstruction. Consensus reads used in the sequence alignments were taken as majority rule, where a nucleotide at a specific site was assigned when it comprised the majority of the nucleotides. As we eliminated any reads with stop codons and we set our PHRED quality score cutoff at 20, which equates to a 1% error rate, it is most likely that variation was due to intra-host virus diversity and not from sequencing errors. Very few SNPs were present, and even fewer were non-synonymous substitutions, so it is unlikely that these resulted from co-infection with different strains of the same virus and more likely resulted from virus replication error within the host.

·     Tables 3,4: It would be useful to know the number of distinct, nonoverlapping reads obtained for each family.

**We have modified included the total number of unique contigs in parentheses after the total reads by family column, including the number of reads that did not form any consensus sequences. We have also combined Tables 3 & 4 into one table to be able to present the number of distinct, nonoverlapping reads for each family.

·     Results: The presentation of results for each detected virus family presumably covers reads from both the urine and fecal samples combined. This should be clearly stated. 

**In each result section for each virus, we have included whether the reads were generated from feces, urine or both in each virus results section.

·     In Supplementary figure 2, the sequence datasets used for phylogenetic reconstruction of the flavivirus E and NS5 genes are different. In some cases, the gene sequences are extracted from whole genome sequences present in GenBank, and in others, they are derived from genome segments present in GenBank. But many of the genome segment sequences have whole genome equivalents present. For example, AF013394 (NS5, 1,011 nt) is 99% similar to NC_034007 (used for E), which is a whole genome sequence. The problem is that this makes it difficult to compare the two trees and detect incongruent relationships between each inferred phylogeny. This is true of all of the 2-tree figures as well. Perhaps the sequences derived from the same (or nearly the same) virus isolate could be labeled with a common identifier. In other cases, the gene segment corresponding to the same isolate was not used in constructing the tree. For Figure 2, rotavirus, the G33P rotavirus isolate sequence for VP1 (KX814935) was used in the alignment. The sequence for the same isolate representing the VP2 sequence was not included in the VP2 tree, even though it is available in GenBank (KX814936). The polytomy described for the VP2 tree (line 233) may very well have been resolved with the inclusion of KX814936. To the extent possible, sequences representing the same isolates should be used in reconstructing the phylogeny for two separate genes of the same virus species.

**We thank the reviewer for their thoughtful comment. Our aim here is not to elucidate the detailed evolutionary history of each of the viruses identified from the deep sequencing. Rather, we are concerned with confirming the Diamond and BlastX assignments through evolutionary relationships with available data. While we understand the reviewers concern, it is not possible to base these analyses on full genomes because so few of these actually exist for these viruses, which is especially true of bat paramyxovirus nucleocapsid sequences. Using the flavivirus trees as an example, the Eonycteris viruses associate with genes (E and NS5) from the Batu Cave and Phnom Penh viruses, demonstrating consistent evolutionary relationships for the 2 genes. The rest of the virus taxa in the tree provide context of the position of the novel virus genes within the overall diversity of flaviviruses. Furthermore, some of the viruses (i.e. Orthoeoviruses and Rotaviruses) have segmented genomes and it make little sense to use the same taxa in analyses for different genes as they will have markedly different evolutionary histories through reassortment.

·      Lines 227-235. This paragraph references the rotavirus VP7 gene when it appears that the reference should be to the VP2 gene. It also references figure 4, when it should reference figure 2.

**Line 982: Our apologies for this mislabelling. This rotavirus phylogenetic tree is for the VP7 gene and not the VP2 gene. All references to VP2 have been changed to VP7.

·     Figure 3, paramyxovirus trees: Why did the Eonycteris spelaea paramyxovirus sequences for NP contain “Nipah” in the label, and why were the L sequences differently labeled? Is there evidence that the sequence reads for these two paramyxovirus genes come from different species? 

**Our apologies for this label. We understand this is not Nipah henipavirus and we have relabelled the Eonycteris Paramyxovirus NP to remove this moniker.

·     Line 307: The phrase “but not limited to other virus families” is a bit awkward.

**We agree with the reviewer that this is awkward. We have re-worded this sentence to read : Line 1206: The most common invertebrate viruses were dicistroviruses (65,353 reads), but there were several other virus families present

·     Discussion, line 346: The sentence beginning “Cross-species transmission is difficult…” is poorly worded and the mention of the human immune system seems out of place.

**We appreciate the comments on this section and have removed these sentences to improve the clarity. We have removed: Cross-species transmission is difficult, as viruses must survive outside the host until encountering susceptible hosts, and be capable of evading the human immune system and infecting the cells [85]. The infidelity of RNA polymerases provides an excellent opportunity for producing novel emerging variants with the potential to infect a wider host range [86]. A number of spill-over outbreaks of bat coronaviruses, filoviruses, henipaviruses, lyssaviruses and reoviruses have been reported in humans [87].

**Lines 1324-1334: now read: We reconstructed individual gene phylogenies for each of the above-mentioned virus families. In most phylogenies, our novel viral sequences of Eonycteris spelaea are clustered within a group of bat viruses from other genera, often from the same family, indicating circulation of these viruses among different bat species. Detection of novel viruses may not provide direct information on zoonotic capacity, but the generated sequence data can allow us to better understand the evolutionary history of these virus families and infer potential cross-species transmission [88]. For instance, our VP1 phylogeny of parvovirus indicates that bat parvoviruses are sister to murine and porcine parvoviruses, whereas the VP2 phylogeny shows bat parvoviruses are closely related to canine, feline and porcine parvoviruses. It is apparent that parvovirus is capable of infecting a broad host species, although further research is needed to understand how the virus jumps to different hosts.

·     Line 360: The paragraph that begins here needs to be completely rethought and rewritten. It contains a mix of ideas and statements that may not necessarily provide the best explanations for the results obtained.

**Lines 1345-1361: We have re-written and re-organized the discussion to clarify the order of operations. Specifically, we have changed this paragraph to note the challenges of conventional PCR vs NGS, how NGS captures virus data, and how our approach removes non-coding sequences. We then discuss our specific results (few virus reads) and the reasons why this may occur.

·     Line 381: The sentence beginning “As this species often exists in very large roosts…” is difficult to comprehend.

**Line 1367: We agree with the reviewer that this needs to be modified. We have changed it to read: ‘As this species roosts in large numbers,’

Reviewer 2 Report

The manuscript by Mendenhall et al. is a nice characterisation of the diversity of viruses found in faecal and urine samples of the bat species Eonycteris spelaea. The virome of this species has not previously been characterised and considering the ecology of this species, which co-roosts with other bat species, it represents and interesting addition to our current knowledge of the diversity of viruses found in bats and the potential transmission of viruses between bat species.

However, the authors have not provided accession numbers for European Nucleotide Archive or Sequence Read Archive, which makes it impossible to reproduce their work or for other researchers to build upon it. Additionally, it will be necessary for the authors to provide the alignments they used for the phylogenies to make their work truly reproducible.

A number of minor points also need to be address:

Line 34: had => has

Line 78: was => were

Citation style changes from [] => ()

Line 101: use of NCBI, line 107: National Center for Biotechnology Information

Line 307: but not limited to other virus families => the sentence is unclear.

Line 348: evading the human immune system => evading the novel host immune system 

Line 362: The sample size of 4 with one single samples sequenced by HiSeq is too small to make the following statement in the discussion: “Notably, more viral reads are generated from the MiSeq Illumina than the HiSeq Illumina.”

Author Response

We would like to thank both reviewer for taking the time to review our manuscript and provide comments and suggestions to improve the quality. We have provided specific responses to each reviewer below. Reviewer one suggested that we provide the number of consensus sequences assembled from individual reads for each virus family. We revisited the NGS datasets and determined the number of consensus reads and unassembled reads for each virus. This lead us to combine tables 3 and 4 into one table that contained the virus families for feces and urine with the number of assembled reads as the last column. For trimming the data, we excluded any X174 phage reads as they likely originated from the library preparation kit. While reviewing the viruses by family in MEGAN from the DIAMOND file, it appeared that there were other bacteriophage sequences that had been removed during the analysis. We have included these reads in the table (families Microviridae, Myoviridae, and Podoviridae). Additionally, the reads from another bacteriophage family (Siphonviridae) were underestimated. These have been added to Table 3 (number of reads). While it does change the proportion of reads and reduces the percentage of reads from dicistroviruses (arthropod-specific viruses), the number of vertebrate-specific viruses remains unchanged. We apologize for this oversight and have remedied it in this revised version of the draft. We have uploaded our NGS fastq files to NCBI’s Sequence Read Archive so they will be publicly available. We have submitted our sequences used in the tree to Genbank and are awaiting accession numbers. Once these are accepted, we will add these to the manuscript and change the identifiers in the phylogenetic tree. Please note that our line references are based on the revised manuscript with changes being tracked.

The manuscript by Mendenhall et al. is a nice characterisation of the diversity of viruses found in faecal and urine samples of the bat species Eonycteris spelaea. The virome of this species has not previously been characterised and considering the ecology of this species, which co-roosts with other bat species, it represents and interesting addition to our current knowledge of the diversity of viruses found in bats and the potential transmission of viruses between bat species.However, the authors have not provided accession numbers for European Nucleotide Archive or Sequence Read Archive, which makes it impossible to reproduce their work or for other researchers to build upon it. Additionally, it will be necessary for the authors to provide the alignments they used for the phylogenies to make their work truly reproducible.

We have addressed the specific comments below with our comments prefaced by ** and in blue

**We thank the reviewer for their comments and constructive instructions to improve our manuscript. We have addressed the minor points below. We agree with the reviewers that the sequence data must be available online. We have submitted each sequence used in the alignments and phylogenies to genbank at NCBI and we have submitted the next generation sequencing datasets to the Sequence Read Archive (SRA) at the National Center for Biotechnology Information (bioProjectID PRJNA524946). This is noted in lines 506-507. The accession numbers have been added to each virus results section and to each sequence name in each phylogenetic tree. All sequence data will be publicly available.

A number of minor points also need to be address:

Line 34: had => has

** ‘Had’ has been changed to ‘has’

Line 78: was => were

** ‘Was’ was changed to ‘were’

Citation style changes from [] => ()

**The citation style has been changed from [] to ()

Line 101: use of NCBI, line 107: National Center for Biotechnology Information

** NCBI has been changed to National Center for Biotechnology Information

Line 307: but not limited to other virus families => the sentence is unclear.

** We have modified this sentence to read: ‘The most common invertebrate viruses were dicistroviruses (65,353 reads), but there were several other virus families present…’

Line 348: evading the human immune system => evading the novel host immune system 

**We thank the reviewer for this comment. After reviewing the comments from review 1 and your suggestion, we have eliminated the following section of the paragraph: Cross-species transmission is difficult, as viruses must survive outside the host until encountering susceptible hosts, and be capable of evading the human immune system and infecting the cells [85]. The infidelity of RNA polymerases provides an excellent opportunity for producing novel emerging variants with the potential to infect a wider host range [86]. A number of spill-over outbreaks of bat coronaviruses, filoviruses, henipaviruses, lyssaviruses and reoviruses have been reported in humans [87].

Line 362: The sample size of 4 with one single samples sequenced by HiSeq is too small to make the following statement in the discussion: “Notably, more viral reads are generated from the MiSeq Illumina than the HiSeq Illumina.”

**We have changed this apply specifically to our study: ‘Interestingly, in our study, more viral reads were generated from the MiSeq Illumina than the HiSeq Illumina.’

Round  2

Reviewer 1 Report

Genus names in Table 1 need to be italicized.

The terms "Global" and "Subsampled" need to be defined, e.g., does  "global" refer the the full length genome sequence, or the full length gene sequence? 

Table 1 needs footnotes to explain the column headers.

Table 3 was truncated in the pdf file, so it could not be reviewed.

Author Response

We thank the reviewer for rapidly reviewing the manuscript. Please find our responses below

Genus names in Table 1 need to be italicized.

**These names have been italcicized

The terms "Global" and "Subsampled" need to be defined, e.g., does  "global" refer the the full length genome sequence, or the full length gene sequence? 

We have changed the phrase to be, " Representative nucleotide sequences specific to the gene of interest were downloaded from the NCBI GenBank...". In this way we remove the confusion about the word global.

 For the phrase, "...the final datasets were further down sampled to reduce redundant and similar sequences.", we believe that we have defined down sampled by following it use with "reduce redundant and similar sequences."

Table 1 needs footnotes to explain the column headers.

We have removed the word Global and used "Initial" and "down sampled" in place of Global and Subsampled.

Table 3 was truncated in the pdf file, so it could not be reviewed.

Our apologies for the formatting error. We have placed the table into its own section and made it landscape form. We hope this improves the clarity.